# Physiotherapy and associated factors affecting mouth opening changes in noma patients during initial hospitalization at an MSF-supported hospital in Northwest Nigeria —A retrospective cohort study

**Oluwole Victor Oluwalomola**[1]*, **Emily Briskin**[1,2], **Michael Olaleye**[1], **Joseph Samuel**[1], **Bukola Oluyide**[3], **Mark Sherlock**[3], **Adeniyi Semiyu Adetunji**[4], **Mohana Amirtharajah**[3]

1 Medecins Sans Frontieres, Sokoto, Nigeria, 2 Medecins Sans Frontieres, Epicentre, Paris, France, 3 Medecins Sans Frontieres, Amsterdam, Netherlands, 4 Department of Surgery, Noma Children Hospital, Sokoto, Nigeria

* oluwolevctr@gmail.com

## Abstract

Noma is a rapidly progressing infection of the oral cavity, which can cause the disintegration of the cheek, nose and eye, in under a week. One of the most disabling sequelae is trismus, the restriction of mouth opening, which results in difficulties in speech, mastication, social feeding habits and maintenance of oral hygiene. Restriction of mouth opening among noma patients mostly begins during the transition between World Health Organisation (WHO) stage 3 (gangrene) and stage 4 (scarring) of the disease. This study aims to describe the impact of physiotherapy in noma patients hospitalised with stages 3 and 4 of the disease and to identify factors that influence change in mouth opening of noma patients. This study is a retrospective analysis of routinely collected data from patients admitted at Noma Children Hospital, Sokoto, Northwest Nigeria between 1 May 2018 and 1 May 2020. Eligible patients included stage 3 and 4 noma patients who had not undergone any surgical reconstruction or trismus release surgery but received physiotherapy assessment and treatment during initial hospitalization. Factors associated with a change in mouth opening were identified using paired t-test analysis, bivariate and multivariate analyses. The mean difference in the mouth opening from admission to discharge was 6.9mm (95% CI: 5.4 to 8.3, p < 0.0001). Increased number of physiotherapy sessions and patient age above three years were significant predictors of improvement in mouth opening (p-value 0.011, 0.001 respectively). Physiotherapy treatment received within an adequate number of physiotherapy sessions for stage 3 and 4 noma patients during the period of the first hospitalization is important and results in a significant increase in mouth opening. Hence, noma patients at these stages should routinely undergo physiotherapy as part of a holistic approach to treatment.

**Data Availability Statement:** All relevant data are within the manuscript and its Supporting Information files.

**Funding:** The authors received no specific funding for this work.

**Competing interests:** The authors have declared that no competing interests exist.

## Introduction

Noma is a rapidly progressing infection of the oral cavity, which can cause the disintegration of the cheek, lips, nose and/or the eye in under a week [1]. Without treatment, noma has a reported 90% mortality rate [2]. The World Health Organization (WHO) classifies noma into six stages: Stage 0: simple gingivitis, Stage 1: acute necrotizing gingivitis, Stage 2: oedema, Stage 3: gangrene, Stage 4: scarring and Stage 5: sequelae [2]. Noma is preventable [3], and mostly affects children caught in the vicious cycle of extreme poverty, chronic malnutrition, a markedly impaired immune system and rising vulnerability to endemic bacterial and viral infections [4]. Treatment in the earlier acute and semi-acute stages of the disease with antibiotics, wound debridement and nutritional support can greatly reduce morbidity and mortality. However, affected patients are still left with significant sequelae including disfigurement, functional impairment, and the inability to move the jaw [2,5,6]. Specifically, noma survivors are often left with incontinentia oris, trismus, interference with the growth of the facial skeleton, deformed dentition, as well as facial deformities that cause difficulty with eating, speaking and appearance, all of which can lead to stigmatization and isolation [3,7]. One of the most disabling sequelae of patients that survive acute noma is trismus, a restriction of mouth opening [8,9], which results in difficulties in speech, mastication, social feeding habits and maintenance of oral hygiene [7].

Restriction of mouth opening among noma patients mostly begins during the transition between WHO stage 3 (gangrene) and stage 4 (scarring) of the disease. The development of fibrous tissue and a reflex muscle spasm of the mouth closing muscles, including the temporalis, masseter and pterygoideus medialis, at the early inflammatory stage, are insidious occurrences during the healing of noma [9]. If nothing is done to combat or ameliorate this process, severe restriction of mouth opening often occurs. Prompt commencement of physiotherapy during this period could prevent severe restriction of mouth opening [5].

The nose, outer lining, inner lining, trismus, lower lip, upper lip, particularities (NOITULP) classification system of noma captures the degree of trismus in noma patients and survivors [8]. The degree of trismus ranges through normal mouth opening T0($\geq$40mm), T1(20-40mm), T2($>$0 up to 20mm) and no mouth opening T3(0mm) [10,11]. However, Marck and Bos [10] also noted that children below the age of 8 years have a normal mouth opening of less than 40mm. As the disease affects children mostly between zero and six years of age, any intervention at improving mouth opening of stage 3 and 4 noma patients should aim at achieving a mouth opening of at least within the T1 degree of trismus (20mm up to 40mm).

Most of the published literature on noma is focused on the outcomes of patients with the sequelae of noma (Stage 5) who are treated with surgical reconstruction [7,10,12–15]. Prior studies have emphasised the need for post-operative physiotherapy for noma patients who undergo trismus release surgery, [7] and better mouth opening has been recorded among those who continued with physiotherapy [11]. However, little or no evidence has described the role of physiotherapy in noma patients at stages 3 & 4 to prevent severe restriction of mouth opening and to improve available mouth opening during the time of initial hospitalization.

Since 2015, MSF has been supporting the Noma Children Hospital, Sokoto, Northwest Nigeria, in providing treatment for noma patients, raising awareness of noma and providing reconstructive surgery for noma survivors. As part of MSF's routine care, stage 3 and 4 noma patients admitted in the hospital receive physiotherapy as a preventive and therapeutic measure to improve mouth opening. The treatment techniques involve performing active and passive mobilization of the temporomandibular joint with the application of stretching techniques on the soft tissues. Wooden tongue depressors stacked between the teeth are used as a therapeutic modality to give sustained stretch to prevent strictures that could arise due to scar contraction.

This study aims to describe the impact of physiotherapy in noma patients hospitalised with stages 3 and 4 of the disease and to identify factors that influence change in mouth opening of noma patients.

## Methodology

### Study design and setting

We conducted a retrospective analysis of routinely collected data of patients admitted to the Noma Children Hospital in Sokoto, Northwest Nigeria from 1 May 2018 to 1 May 2020. The hospital was established in 1999 and has been receiving support from MSF since 2015. A full-time physiotherapy service in the hospital began in late 2017, hence noma patients' mouth opening data before this time were not available. The hospital treats an average of 100 cases of noma per year.

### Study participants

WHO Stage 3 (gangrene) and 4 (scarring) noma patients who were admitted at the Noma Children Hospital during the study period that have not undergone any surgical reconstruction or trismus release surgery but received physiotherapy assessment and treatment during initial hospitalization were included in the study. Patients routinely collected data were used retrospectively in this study and as such patients were neither approached nor assessed for research purposes. Age as one of the variables in this study was categorized into two groups which are age group 0-3yrs and age group >3yrs based on the approximate age at which a child can independently understand and respond to instructions during a physiotherapy session.

### Data collection

Relevant variables from the database were exported into a password-protected excel-based spreadsheet. These variables included patients' sociodemographic characteristics, mouth opening measurements and the time points at which they were taken, duration of hospital stay, number of physio sessions received, caregivers' relationship to the patient, defect site, and whether sequestrum removal was done. The data were anonymised and only linked to the main noma database through unique patient identifying numbers.

### Statistical analysis

A paired t-test was used to evaluate the difference between mouth opening at admission and discharge from the first hospital visit. Categorization of individual patients' mouth opening at admission and discharge into two groups (<20mm & ≥20mm) was done, using 20mm as the benchmark for acceptable functional mouth opening in this age group. The percentage of patients in the different groups at these time points was ascertained.

Bivariate analyses (one-way ANOVA analysis and linear regression analysis) were performed to assess which factors were associated with a change in mouth opening using a confidence interval of 95% and p-value < 0.05.

All the determinants with a p-value less than 0.05 in the bivariate analyses were used in the multivariate analysis. All assumptions to be met for linear regression and multivariate analysis were checked and tested. Two determinants (number of physiotherapy sessions and length of hospital stay) were discovered to have multicollinearity, so the length of hospital stay was excluded from the multivariate analysis. A multivariate linear regression analysis was conducted to assess the relationship between patient age, number of physiotherapy sessions, and

the dependent variable, change in mouth opening. All data were analysed using R Core Team (2020). R: A language and environment for statistical computing. (R Foundation for Statistical Computing, Vienna, Austria. URL https://www.R-project.org/.)

## Ethical considerations

This research fulfilled the exemption criteria set by the MSF Ethics Review Board (ERB) and the National Health Research Ethics Committee, Federal Ministry of Health, Nigeria for a posteriori analysis of routinely collected clinical data. It was conducted with permission from the Medical Director of the MSF Operational Centre Amsterdam. Ethical approval was also sought for and approved by the Usmanu Danfodiyo University Teaching Hospital (UDUTH) Health Research and Ethics Committee in Sokoto, Nigeria, the Sokoto Ministry of Health, Ethics Departments. Consent was not sought for this study as this is a retrospective review of routinely collected data and was deemed to be of minimal risk to participants by two ethical review boards.

## Results

### Demographics

In total, 91 patients were included in the analysis; 47 (51.1%) were female and the mean age was 5 years (Std 5). Most (72/91, 78.3%) patients had a cheek defect upon admission. Twenty-six (28.3%) patients had sequestrectomy done during initial hospitalization. The mean number of physiotherapy sessions was 11 (Std 6), and the mean length of hospital stay were 25 days (Std 16). (Table 1).

**Mouth opening changes.** Patients had a mean mouth opening of 17.6mm at admission and 24.5mm at discharge. The mean difference between the time points was 6.9mm (95% CI: 5.4 to 8.3; paired t-test: t = 9.4, df = 91, p = <0.001) "Fig 1".

**Table 1. Sociodemographic and clinical characteristics of noma patients at the MSF supported "Noma Children Hospital" in Sokoto, Nigeria between May 2018-May 2020.**

| Characteristic | | | Frequency | Percentage | Mean (Standard deviation) |
|---|---|---|---|---|---|
| *Age (years) (n = 90)* | | | | | 5(5) |
| | Age group | 0–3 | 44 | 47.8% | |
| | | >3 | 46 | 50.0% | |
| *Number of physiotherapy sessions completed* | | -- | -- | -- | 11(6) |
| *Sex (n = 91)* | | | | | -- |
| | | Male | 44 | 47.8% | |
| | | Female | 47 | 51.1% | |
| *Length of Hospital stay (days)* | | -- | -- | -- | 25(16) |
| *Caregiver relationship* | | | | | -- |
| | | Primary caregiver | 72 | 78.3% | |
| | | Not primary caregiver | 8 | 8.7% | |
| *Defect Site (n = 91)* | | | | | -- |
| | | Cheek | 72 | 78.3% | |
| | | Lip | 11 | 11.9% | |
| | | Nose | 6 | 6.5% | |
| | | No defect | 2 | 2.2% | |
| *Sequestrectomy (n = 81)* | | | | | -- |
| | | Yes | 26 | 28.3% | |
| | | No | 55 | 59.8% | |

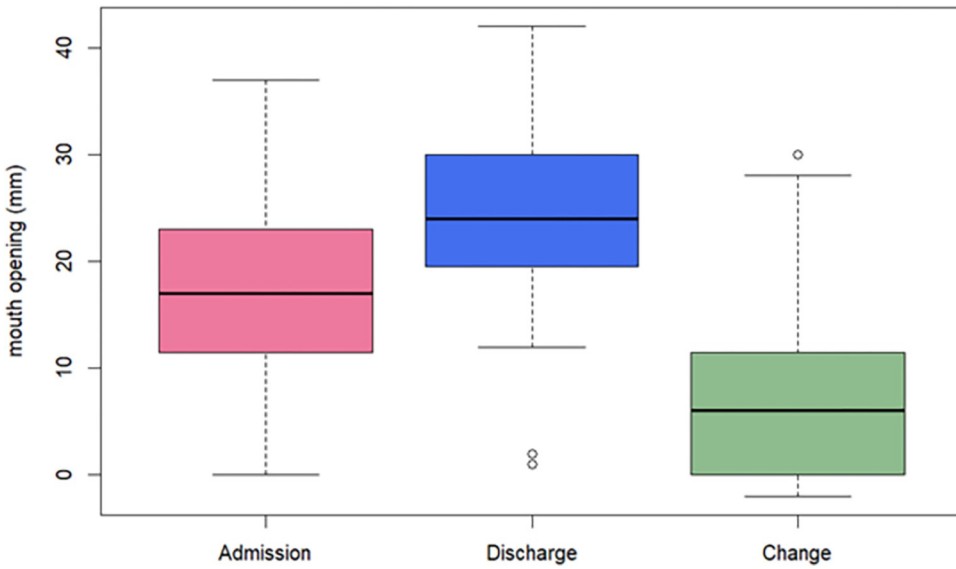

**Fig 1. Box plot showing the comparison of average mouth opening of respondents at admission and discharge (n = 91), and change in mouth opening of noma patients at the MSF supported "Noma Children Hospital" in Sokoto, Nigeria between May 2018-May 2020.**

"Fig 1". Box plot of mouth opening at admission, discharge and change in mouth opening. The bold lines indicate the mean mouth opening (17.6mm at admission and 24.5mm at discharge) and the rectangles indicate the Q1-Q3 interval.

There was a significant difference in the average change in mouth opening for those aged 0–3 years (4.5mm) and those above three years (8.9mm). The difference between the mean mouth opening changes of these two age groups was 4.5mm (95% CI: 1.7 to 7.2; independent t-test: t = 3.2, df = 88, p = 0.002) "Fig 2".

"Fig 2". Box plots of change in mouth opening between different age groups. The bold lines indicate the mean change in mouth opening of age group 0–3 years to be 4.5mm and above 3 years to be 8.9mm.

47 patients (52%) had a <20mm mouth opening while 44 patients (48%) had a > = 20mm on admission. At discharge, this proportion shifted to 23 patients (25%) with a <20mm mouth opening, and 68 patients (75%) with a > = 20mm mouth opening. Thus, 27% (more than a quarter of patients) went from a non-functional to a functional mouth opening Figs 3 and 4.

## Mouth opening distribution at admission

"Fig 3. Pie chart of individual patients' mouth opening categorized into two groups. The yellow-coloured pie indicates the percentage of patients admitted with mouth opening less than 20mm (T2) while the red-coloured pie indicates the percentage of patients admitted with 20mm or more (T3).

## Mouth opening distribution at discharge

"Fig 4. Pie chart of individual patients' mouth opening categorized into two groups. The yellow-coloured pie indicates the percentage of patients discharged with mouth opening less than 20mm (T2) while the red-coloured pie indicates the percentage of patients discharged with 20mm or more (T3).

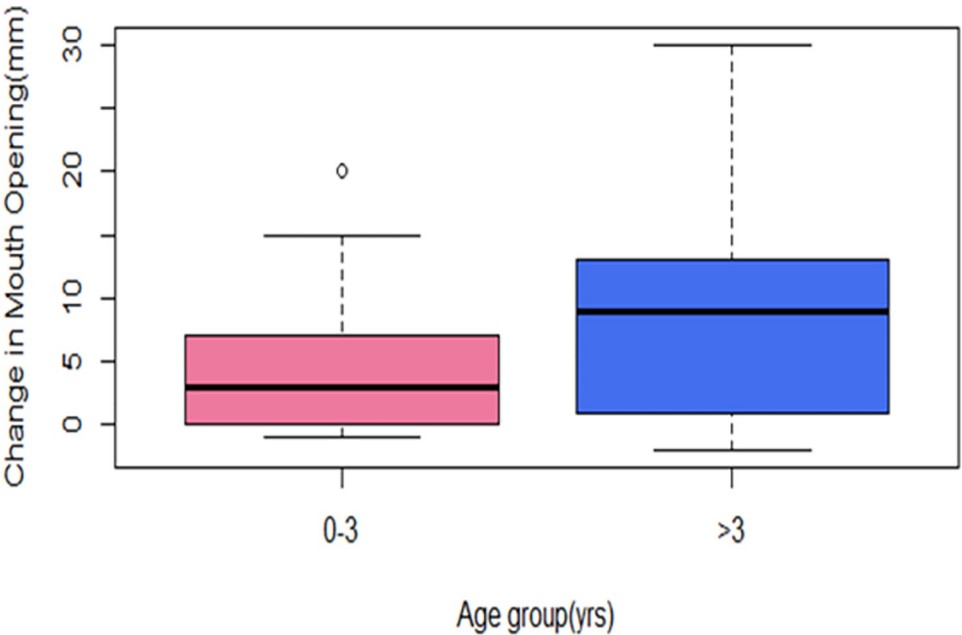

**Fig 2. Box plot comparing the change in mouth opening of different age groups of respondents (0–3 years and above 3 years) (n = 88), of noma patients at the MSF supported "Noma Children Hospital" in Sokoto, Nigeria between May 2018-May 2020.**

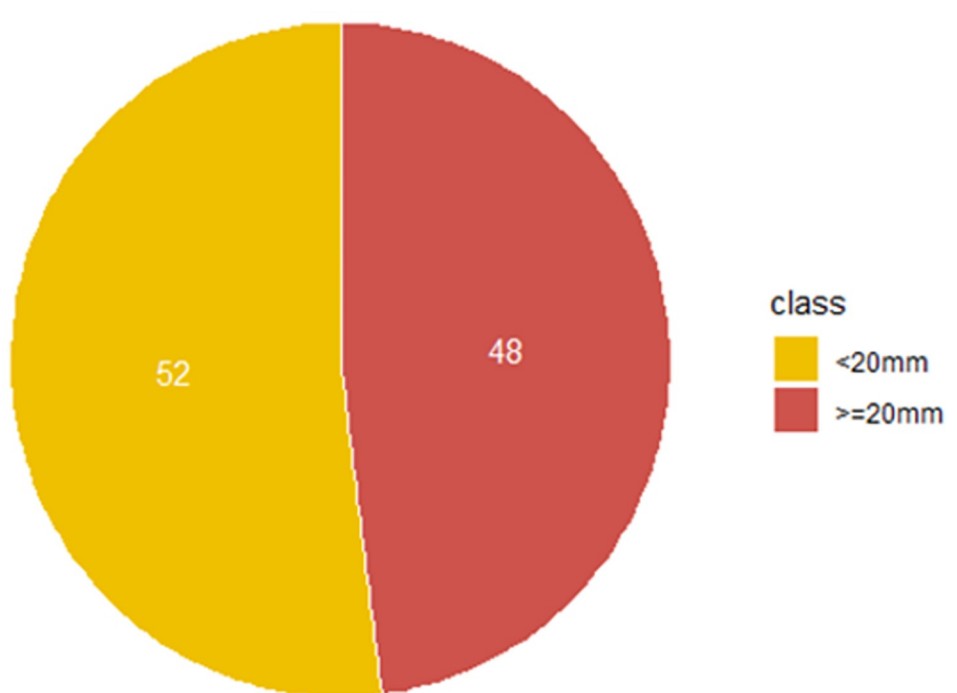

**Fig 3. Proportion of patients with mouth opening below 20mm and 20mm above at point of admission (n = 91), of noma patients at the MSF supported "Noma Children Hospital" in Sokoto, Nigeria between May 2018-May 2020.**

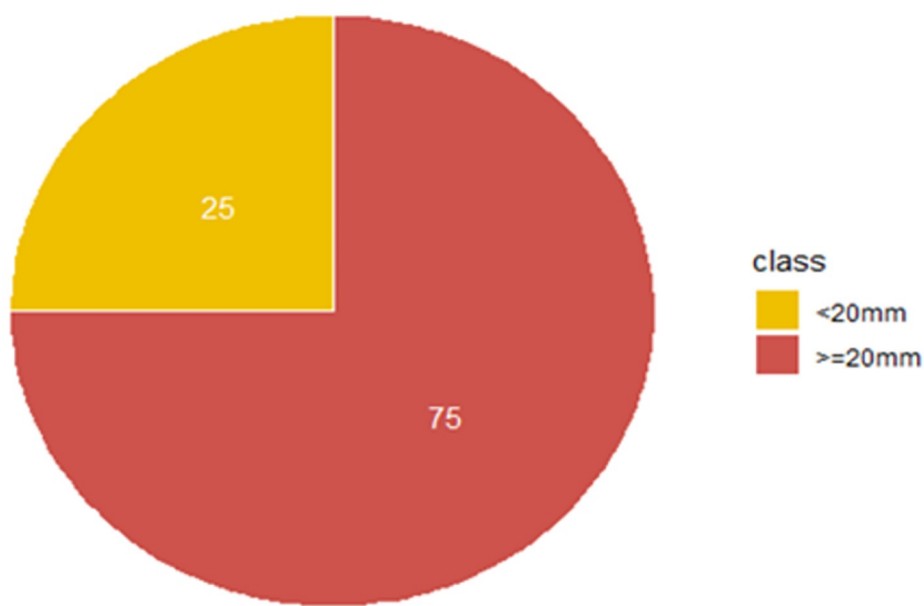

**Fig 4. Proportion of patients with mouth opening below 20mm and 20mm above at point of discharge (n = 91), of noma patients at the MSF supported "Noma Children Hospital" in Sokoto, Nigeria between May 2018-May 2020.**

## Predictive factors for changes in mouth opening

The bivariate analysis showed that age group, number of physio sessions and length of hospital stay were associated with a change in mouth opening measurements (Table 2).

The multivariate analysis showed that an increased number of physiotherapy sessions (p = 0.011) and being aged above three years (p = 0.001) were significant predictors of improvement in mouth opening (Table 3). For each additional physio session, a 0.3mm increase in mouth opening was observed (95% CI: 0.1 to 0.5mm). Being greater than age 3 accounted for a 4.8mm increase in mouth opening in the fully adjusted model (95% CI: 2.2 to 7.5mm).

## Discussion

This study has shown that patients admitted with WHO stages 3 and 4 noma disease in Noma Hospital Sokoto who undergo physiotherapy experience significant improvement in mouth opening from admission to discharge, with increased number of physiotherapy sessions and being aged over three years significantly associated with the improvement in mouth opening.

These findings confirm the positive impact of the inclusion of physiotherapy management in the treatment of noma to prevent restriction and improve mouth opening [5]. This finding is consistent with the positive outcomes of physiotherapy treatment of trismus caused by other diseases, such as head and neck cancer, as reported by Kamstra et al., [16] and Dıraçoğlu et al., [17]. Thus, this study on noma patients supports the use of physiotherapy as part of medical care for trismus regardless of aetiology. This study also confirms that increases in mouth opening were achieved with additional physiotherapy sessions, a finding that makes intuitive sense. However, further studies are needed to establish the optimal number of physiotherapy sessions required to achieve functional mouth opening in these patients before the improvements in opening level off.

**Table 2.  Bivariate analysis of patients characteristics and change in mouth opening (mm) of noma patients at the MSF supported "Noma Children Hospital" in Sokoto, Nigeria between May 2018-May 2020.**

| Characteristic | | Frequency | Percentage | Mean change in mouth opening (mm) | P-value |
|---|---|---|---|---|---|
| *Age (n = 90)* | | | | | 0.002*** |
| | 0–3 | 44 | 47.8% | 4.47 | -- |
| | >3 | 46 | 50.0% | 8.95 | -- |
| *Sex (n = 91)* | | | | | 0.89 |
| | Male | 44 | 47.8% | 6.97 | -- |
| | Female | 47 | 51.1% | 6.78 | -- |
| *Caregiver relationship (n = 80)* | | | | | 0.75 |
| | Primary caregiver | 72 | 78.3% | 6.83 | -- |
| | Not primary caregiver | 8 | 8.7% | 6.00 | -- |
| *Defect Site (n = 91)* | | | | | 0.73 |
| | Cheek | 72 | 78.3% | 7.22 | -- |
| | Lip | 11 | 11.9% | 6.54 | -- |
| | Nose | 6 | 6.5% | 4.33 | -- |
| | No defect | 2 | 2.2% | 4.00 | -- |
| *Sequestrectomy (n = 81)* | | | | | 0.73 |
| | Yes | 26 | 28.3% | 5.92 | -- |
| | No | 55 | 59.8% | 6.49 | -- |
| | | **Coefficients** | **95% Confidence Interval** | | |
| | | | Lower Bound | Upper Bound | |
| *Number of physio sessions completed (n = 20)* | | 0.28 | 0.04 | 0.53 | 0.02** |
| *Length of Hospital stay (n = 75)* | | 0.14 | 0.05 | 0.22 | 0.002*** |

Moreover, this study showed that those aged above three years had greater improvements in mouth opening than younger patients. This improvement is likely attributed, at least in part, to the fact that the cognitive development of children differs between younger and older age groups, allowing older children to play a more active role during physiotherapy sessions [18,19]. Based on physiotherapy staff observation, older children participate, adhere and comply to exercise therapy and the use of therapeutic modalities better than younger ones due to their ability to grasp the purpose or function of the treatment. However, a physiologic cause for the discrepancy in improvement in younger patients cannot be excluded. Regardless of aetiology, to achieve better outcomes in children three years of age and younger, adapted physiotherapy methods may need to be developed.

A normal mouth opening measurement (40mm) might not be attained by noma patients despite receiving physiotherapy treatments due to the loss of elastic properties of facial soft tissues at the defect site. However, the inability of a person with noma to regain normal or full mouth

**Table 3.  Multivariate analysis of study variables and change in mouth opening(mm), of noma patients at the MSF supported "Noma Children Hospital" in Sokoto, Nigeria between May 2018-May 2020.**

| Characteristic | | Coefficients | 95% Confidence Interval | | P-value |
|---|---|---|---|---|---|
| | | B | Lower Bound | Upper Bound | |
| | (Constant) | 0.9 | -2.6 | 4.1 | 0.589 |
| | Number of physiotherapy Sessions | 0.3 | 0.1 | 0.5 | 0.011*** |
| | Age group >3 | 4.8 | 2.2 | 7.5 | 0.001*** |

opening is not always a serious clinical problem as an opening of less than 40mm can still be functional [10]. Nevertheless, the focus should be on achieving a mouth opening of ≥20mm (T1) for these patients, which is functional enough for oral hygiene and adequate feeding. Study participants had a mean mouth opening of 24.5mm at discharge after receiving physiotherapy. This amount of mouth opening is not associated with any serious oral incapability compared to a <20mm (T2) mouth opening which could pose a clinical or functional problem [10]. This study also revealed that of the 91 patients that participated in this study, only 23 patients (25%) were discharged with a mouth opening of <20mm, which is in contrast to 47 patients (52%) with <20mm mouth opening at admission. Although those who were discharged with <20mm of mouth opening also showed some improvement with physiotherapy, this improvement did not reach the amount of opening that is needed to prevent functional difficulties.

The length of hospital stay influences how many physiotherapy sessions patients can partake in before being discharged. Patients with longer hospital stay on average have more physiotherapy sessions, which our study has shown to be associated with improvement of mouth opening. Facilities treating noma patients may want to modify discharge criteria to account for these findings, by either extending the length of the hospital stay for certain patients to support the achievement of a functional mouth opening, or offering targeted outpatient physiotherapy support to certain patients to attain functional mouth opening after discharge.

Caregivers' involvement in health interventions for children plays an important role [20], hence the need to involve caregivers in supporting noma patients during physiotherapy sessions as this might affect the desired outcome (improvement in mouth opening). However, it is unclear if the relationship of the caregiver affects a change in mouth opening. Our study did not identify any association between the relationship of caregivers to the noma patient that received physiotherapy treatment and change in mouth opening, though the complex caregiving relationships may have been difficult to categorize and analyse through this quantitative analysis and needs further exploration.

The study did have some limitations. As a retrospective study of routinely collected data, the number of variables that could be analysed for their relationship to mouth opening was limited to what was included in the database. For example, the presence of malnutrition or other co-morbidities may affect the gains made in physiotherapy but were not evaluated in this study. In addition, this study did not contain a comparison group of patients who did not undergo physiotherapy. However, given that trismus is a debilitating condition, the withholding of physiotherapy for patients potentially at risk for trismus would not be ethical; thus, all patients at risk of trismus admitted to NCH undergo physiotherapy as part of their routine care. Finally, further studies are needed to determine the minimum number of physiotherapy sessions needed to improve trismus in acute noma patients as well as to determine the best methods of achieving functional mouth opening in patients 3 years of age and younger.

## Conclusion

To date, no studies have identified or described the impact of physiotherapy on mouth opening in stage 3 and 4 noma patients prior to surgery. This study shows that the inclusion of physiotherapy in stage 3 and 4 noma patients during the period of the first hospitalization is a crucial part of care. Special attention should be given to developing physiotherapy methods for noma patients under 3 years of age, and facilities should consider models of care that may include physiotherapy and mouth opening benchmarks among discharge criteria. Physiotherapy should be added to the package of standard best practices as part of a holistic approach to the treatment of stage 3 and 4 noma.

## Supporting information

**S1 Data.**
(XLSX)

## Acknowledgments

We sincerely appreciate the staff at MSF-OCA, Nigeria Mission and the Noma Children Hospital, Sokoto that has contributed in one way or the other to the success of this study. We say a big thank you to Elise Farley for her immense contribution, support and guidance. Thanks to Elburg Van Botzlear for critically reviewing the early versions of this manuscript and Emily Amundson for making out time out of her busy schedule to review this manuscript.

## Author Contributions

**Conceptualization:** Oluwole Victor Oluwalomola, Emily Briskin.

**Data curation:** Michael Olaleye.

**Formal analysis:** Oluwole Victor Oluwalomola.

**Methodology:** Oluwole Victor Oluwalomola, Emily Briskin, Mohana Amirtharajah.

**Resources:** Joseph Samuel, Bukola Oluyide.

**Supervision:** Mohana Amirtharajah.

**Validation:** Mark Sherlock, Mohana Amirtharajah.

**Visualization:** Oluwole Victor Oluwalomola, Emily Briskin, Michael Olaleye.

**Writing – original draft:** Oluwole Victor Oluwalomola, Emily Briskin.

**Writing – review & editing:** Joseph Samuel, Bukola Oluyide, Mark Sherlock, Adeniyi Semiyu Adetunji, Mohana Amirtharajah.

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
