## [Decision Letter · Decision Letter 0]

11 Apr 2023

PGPH-D-22-01615

Physiotherapy and associated factors affecting mouth opening changes in noma patients during initial hospitalization at an MSF- supported hospital in Northwest Nigeria.

Dear Dr. Oluwalomola,

Thank you for submitting your manuscript to PLOS Global Public Health. After careful consideration, we feel that it has merit but does not fully meet PLOS Global Public Health’s publication criteria as it currently stands. Therefore, we invite you to submit a revised version of the manuscript that addresses the points raised during the review process.

We look forward to receiving your revised manuscript.

Kind regards,

Javier H Eslava-Schmalbach, M.D., Ph.D., MSc

Academic Editor

Journal Requirements:

Additional Editor Comments (if provided):

We have finally received the reviewers' comments. Even though they suggested that we accept your manuscript, we would like you to make some improvements. Specifically, please:

1. Add a statement mentioning the normal distribution of the dependent variable;

2. Clarify why you sometimes use the median and other times the mean to describe the behavior of the data (Table 1 and the following paragraph);

3. Mention if other assumptions needed to run linear regression models were tested and;

4. Introduce the meaning of the acronyms when you mention them for the first time."

We hope that you will make these changes and send them to us as soon as possible

Reviewers' comments:

Reviewer's Responses to Questions

**Comments to the Author**

1. Does this manuscript meet PLOS Global Public Health’s publication criteria? Is the manuscript technically sound, and do the data support the conclusions? The manuscript must describe methodologically and ethically rigorous research with conclusions that are appropriately drawn based on the data presented.

Reviewer #1: Yes

Reviewer #2: Yes

2. Has the statistical analysis been performed appropriately and rigorously?

Reviewer #1: Yes

Reviewer #2: Yes

3. Have the authors made all data underlying the findings in their manuscript fully available (please refer to the Data Availability Statement at the start of the manuscript PDF file)?

Reviewer #1: Yes

Reviewer #2: Yes

4. Is the manuscript presented in an intelligible fashion and written in standard English?

Reviewer #1: Yes

Reviewer #2: Yes

5. Review Comments to the Author

Reviewer #1: The authors report on the positive impact of physiotherapy on improving mouth opening in noma patients. They describe the pathophysiology of the restriction of mouth opening in these patients. The description of the role of physiotherapy in Stages 3 & 4 is unique and important. The full-time physiotherapy service at the Noma Children’s Hospital facilitates the efforts to improve the mouth opening of these children. The opportunity to improve the mouth opening of these young patients can significantly impact the quality of their lives and relieve suffering. Thank you very much for your dedication to helping noma patients, particularly doing the research and publishing your findings. The article is well written and uniquely addresses one of the major problems for these most vulnerable children affected by noma.

Reviewer #2: This manuscript is according to PLOS Global Public Health’s publication criteria. I think this research is a very good approximation about the impact of physioteray in NOMA patients and the identifcation of factors that can influence change in mouth opening of these patients because there is no evidence available related to this topic and the data support strongly the conclusions. It opens the doors to research and establishment of treatment policies for these patients, aimed at improving their health and quality of life. Of course, being a retrospective study, it has limitations such as the existence of possible confounding variables that are recognized by the authors in the manuscript and clearly explained

I think it is very important that the authors are clear about the scope and limitations of the research to develop prospective studies from them, which allow to establish specifically the characteristics of the physiotherapeutic treatment that children should receive, according to their age, nutritional status, hospitalization stay and development of the disease. In addition, it is interesting that the need for physiotherapeutic treatment even after hospital discharge arises. This research is a first step in what can be a line of research in the treatment of physiotherapy of NOMA patients and of much greater impact if it is an interdisciplinary approach, which will clearly improve the health and quality of life of children

6. PLOS authors have the option to publish the peer review history of their article (what does this mean?). If published, this will include your full peer review and any attached files.

**Do you want your identity to be public for this peer review?** For information about this choice, including consent withdrawal, please see our Privacy Policy.

Reviewer #1: **Yes: **M. Leila Srour

Reviewer #2: **Yes: **Erica Mabel Mancera Soto

---

## [Author Response · Author response to Decision Letter 0]

9 May 2023

Thank you for your esteemed feedback about this study. 

We have been able to include in the study the limitations of this research work as requested by the reviewer. We have also effected necessary changes as requested by the editor and these changes have been highlighted in the "response to reviewers" file. Thank you for your consideration of this research work for publication.

---

## [Editor Report · Decision Letter 1]

25 May 2023

PGPH-D-22-01615R1

Physiotherapy and associated factors affecting mouth opening changes in noma patients during initial hospitalization at an MSF- supported hospital in Northwest Nigeria.

Dear Dr. Oluwalomola,

Thank you for submitting your manuscript to PLOS Global Public Health. After careful consideration, we feel that it has merit but does not fully meet PLOS Global Public Health’s publication criteria as it currently stands. Therefore, we invite you to submit a revised version of the manuscript that addresses the points raised during the review process.

We look forward to receiving your revised manuscript.

Kind regards,

Javier H Eslava-Schmalbach, M.D., Ph.D., MSc

Academic Editor

Journal Requirements:

2. Please resubmit the article type to Research Article.

Additional Editor Comments (if provided):

Thank you for submitting the new version of your article. There is only one thing that was not mentioned by the reviewers or me, and it should be corrected. Please add the type of study at the end of the title, as suggested by STROBE. Once this change is made, we will accept your manuscript.

Best regards,
---

## [Editor Report · Decision Letter 2]

26 Jul 2023

Physiotherapy and associated factors affecting mouth opening changes in noma patients during initial hospitalization at an MSF- supported hospital in Northwest Nigeria-A Retrospective cohort study

PGPH-D-22-01615R2

Dear Mr Oluwalomola,

We are pleased to inform you that your manuscript 'Physiotherapy and associated factors affecting mouth opening changes in noma patients during initial hospitalization at an MSF- supported hospital in Northwest Nigeria-A Retrospective cohort study' has been provisionally accepted for publication in PLOS Global Public Health.

Best regards,

Javier H Eslava-Schmalbach, M.D., Ph.D., MSc

Academic Editor